

# Estimating the differences in critical thermal maximum and metabolic rate of *Helicoverpa punctigera* (Wallengren) (Lepidoptera: Noctuidae) across life stages

Samuel A. Bawa[1,2], Peter C. Gregg[3], Alice P. Del Soccoro[3], Cara Miller[4] and Nigel R. Andrew[1]

[1] Zoology, Insect Ecology Laboratory, University of New England, Armidale, NSW, Australia
[2] Asuansi Agric. Station, Cape Coast, Central Region, Ghana
[3] Agronomy and Soil Science, University of New England, Armidale, NSW, Australia
[4] Science and Technology, University of New England, Armidale, NSW, Australia

## ABSTRACT

Temperature is a crucial driver of insect activity and physiological processes throughout their life-history, and heat stress may impact life stages (larvae, pupae and adult) in different ways. Using thermolimit respirometry, we assessed the critical thermal maxima ($CT_{max}$-temperature at which an organism loses neuromuscular control), $CO_2$ emission rate ($\acute{V}CO_2$) and Q10 (a measure of $\acute{V}CO_2$ temperature sensitivity) of three different life stages of *Helicoverpa punctigera* (Wallengren) by increasing their temperature exposure from 25 °C to 55 °C at a rate of 0.25 °C min$^{-1}$. We found that the $CT_{max}$ of larvae (49.1 °C ± 0.3 °C) was higher than pupae (47.4 °C ± 0.2 °C) and adults (46.9 °C ± 0.2 °C). The mean mass-specific $CO_2$ emission rate (ml $\acute{V}CO_2$ h$^{-1}$) of larvae (0.26 ± 0.03 ml $\acute{V}CO_2$ h$^{-1}$) was also higher than adults (0.24 ± 0.04 ml $\acute{V}CO_2$ h$^{-1}$) and pupae (0.06 ± 0.02 ml $\acute{V}CO_2$ h$^{-1}$). The $Q_{10}$: 25–35 °C for adults (2.01 ± 0.22) was significantly higher compared to larvae (1.40 ± 0.06) and $Q_{10}$: 35–45 °C for adults (3.42 ± 0.24) was significantly higher compared to larvae (1.95 ± 0.08) and pupae (1.42 ± 0.98) respectively. We have established the upper thermal tolerance of *H. punctigera*, which will lead to a better understanding of the thermal physiology of this species both in its native range, and as a pest species in agricultural systems.

## INTRODUCTION

Insects have a specific range of temperatures within which they can survive, grow and reproduce (*Andrew, Hart & Terblanche, 2011*; *Huang & Li, 2015*; *DeVries, Kells & Appel, 2016*), and have their distribution strongly defined by a thermal envelope (*Gullan & Cranston, 2014*; *Li et al., 2019*). Assessing the impact of extreme temperature exposures on insect distribution is vital to understanding insect ecology and the current and future effects of climate change (*Diffenbaugh et al., 2008*; *Andrew, Hart & Terblanche, 2011*; *Andrew, 2013*; *Betini, Griswold & Norris, 2013*; *Andrew & Hill, 2017*, *Andrew &*

Corresponding author
Samuel A. Bawa,
ambiasin@yahoo.com

*Terblanche, 2013*). Within a rapidly changing climate, thermal limits are an essential part of assessing thermal physiology of insects, as they are the critical endpoints that an organism can survive or loses muscular control (*Lighton & Turner, 2004*; *Andrew, Ghaedi & Groenewald, 2016*), quantified in terms of critical thermal maximum ($CT_{max}$) and critical thermal minimum ($CT_{min}$) (*Kingsolver & Umbanhowar, 2018*).

Thermal limit measurements like $CT_{max}$ and $CT_{min}$ are commonly used to understand and predict species distributions among populations (*Shik et al., 2019*) and responses to environmental changes (*Angilletta et al., 2003*; *Angilletta & Angilletta, 2009*; *Kingsolver & Umbanhowar, 2018*; *Angilletta, Michael & Dunham, 2003*). $CT_{max}$ is often aligned to the distribution of species, as it is a biologically significant and ecologically relevant thermal performance trait (*Sunday et al., 2014*; *Shik et al., 2019*). However, the $CT_{max}$ of most insect groups is highly plastic, so can change depending on an organism's pre-exposure conditions, making it difficult to forecast their response to a rapidly changing climate (*Terblanche et al., 2005*; *Martínez, Cadena & Torres, 2016*; *Kellermann, van Heerwaarden & Sgrò, 2017*).

$CT_{max}$ has been shown to differ across life stages in some insects *e.g.* kelp fly (*Paractora dreuxi*), Sirex wasp (*Sirex noctilio*) and mealworm (*Tenebrio molitor*) (*Marais, Terblanche & Chown, 2009*; *Vorhees & Bradley, 2012*; *Li et al., 2019*), and can be influenced by body mass and surface area of the insect (*Lighton & Turner, 2004*). $CT_{max}$ values depend substantially on how they are measured (*Lighton & Turner, 2004*; *Shik et al., 2019*); it is thus essential to determine a precise scale of temperature exposure for insects (*Terblanche et al., 2007*). Temperature ramping rates can influence the $CT_{max}$ of an insect (*Lighton & Turner, 2004*; *Terblanche et al., 2007*). A slow temperature ramp rate of an organism leads to increased risk of heat shock, which is absent under the natural conditions (*Lutterschmidt & Hutchison, 1997*). The other process that can occur during a slow ramp is short-term physiological adjustments (*e.g.* induction of heat shock proteins) that protect cells from heat damage and might increase $CT_{max}$ (*Kingsolver, 2009*; *Shik et al., 2019*). Fast temperature ramp leads to delay in thermal equilibrium between the air and the insect's body. Because of this, under a fast temperature ramp, the inner temperature of the insect will be lower than the recorded air temperatures leading to an overestimation of their $CT_{max}$ (*Lighton & Turner, 2004*; *Andrew et al., 2013a*; *Agudelo-Cantero & Navas, 2019*).

Intriguingly, only few studies have looked at thermal variation across different life stages (*Le Lann et al., 2011*). The few studies focused on a single unit of fitness in one environment (*Le Lann et al., 2011*). Thermal plasticity has been identified in many insect taxa *e.g.* ants (*Iridomyrmex purpureus*), *Drosophila* species and parasitic wasps (*Aphidius rhopalosiphi*) (*Chown & Terblanche, 2006*; *Le Lann et al., 2011*; *Andrew et al., 2013a*; *Shik et al., 2019*), but not all (*Terblanche, Clusella-Trullas & Chown, 2010*).

Insect herbivores that are found on the ground and plant surfaces in warmer and drier environments are exposed to substantial microclimate variability including seasonal and daily temperature variations at the soil and plant surface (*Hodgson, 1991*; *Bale et al., 2002*; *Andrew, Ghaedi & Groenewald, 2016*). How organisms respond to the variable microclimates depends on the thermal physiology *e.g.* $CT_{max}$ (*Somero, 2010*; *Shik et al., 2019*). The physiology of insects can be modified by the microclimate they experience

(*Andrew et al., 2013a*; *Andrew, Ghaedi & Groenewald, 2016*). Predicting the future in changing climates is made difficult by our limited understanding of the eco-physiological mechanisms and evolutionary process underlying thermal phenotypic plasticity (*Shik et al., 2019*; *Arnold, Nicotra & Kruuk, 2019*).

*Helicoverpa punctigera* is native to Australia and widely distributed across the continent (*Gregg et al., 2016*). It is a notable pest on crops like cotton, oilseeds, legumes and vegetables (*Gregg et al., 2019*). The adult can travel a long distance and can adapt locally to different environmental conditions including inland Australia (*Gregg et al., 2016*, *2019*). Female adult *H. punctigera* lay eggs on leaves, flower buds and developing fruits of the host plant (*Zalucki et al., 1986*). The neonate (newly hatched) usually eat soft leaves; as they mature, they move about other feeding sites like flowers or flower buds, pods, fruits and seeds (*Gregg et al., 2019*). The last larval instar burrows into the soil and pupates. Inland habitats are characterised by extreme climatic conditions like very hot temperatures as high as 50 °C during summer (*Bureau of Meteorology, 2018*). The ability of this insect to survive the extreme summer temperatures in the inland depends on their thermal tolerances and physiological sensitivities *e.g.*, $CT_{max}$ and biochemical shifts that are temperature-dependent (*Gaitán-Espitia et al., 2014*). Female adults of *H. punctigera* are responsible for host selection and dispersal, whilst feeding activities of the larvae cause economic loss to the host plants, and a complete reorganisation of the body occurs within the pupal stage, understanding the metabolic rates among the different life stages are essential.

Energy is the crucial currency of life-history traits of animals including insects (*Terblanche & Chown, 2007*; *Desforges et al., 2019*). Metabolism combines organism's energy supply and fitness-related activities (*Shah et al., 2021*). Measurement of metabolic rate is one way of measuring the energy cost related with different life-functions like growth, development, movement among others (*Shah et al., 2021*). In insects, metabolic rates can influence not only fitness in individuals but also geographic distributions and abundances (*Shah et al., 2021*). Therefore, estimating the metabolic rates of organisms will assist in identifying the cost of living and performance in a particular environment (*Terblanche et al., 2005*; *Terblanche & Chown, 2007*), and can help predict geographic variations in response to global warming (*Shah et al., 2021*).

Metabolic rate can respond to changes in ambient temperature, activity and body mass; however, age and ontogeny, sex, feeding status, season and time of day, among others, can also exert substantial influence (*Terblanche & Chown, 2007*; *Harrison, 2009*; *Altermatt, 2010*). For example, in the Australian field cricket (*Teleogryllus oceanicus*) and Thynnine wasp (*Zaspilothynnus nigripes*), males have a higher metabolic rate than females (*Kolluru, Chappell & Zuk, 2004*; *Tomlinson & Phillips, 2015*). Specific dynamic action is the amount of energy expenditure above the basal metabolic rate (RMR) due to the ingestion and digestion of food for use as energy or conversion to a storage form (*Secor, 2009*). One major metabolic variable is the temporary rise in metabolic rate following feeding status or swallowing food (*Karasov & del Rio, 2007*; *McCue, 2012*). This feeding status-associated rise in metabolic rate is a result of food breakdown and processing in the gut of the organism (*Karasov & del Rio, 2007*). High temperatures generally produce high

metabolic rate, however the total energy assigned for specific dynamic actions is not fully dependent on temperature (*McCue et al., 2016*; *Wang et al., 2002*).

The temperature sensitivity ($Q_{10}$), defined as the rate ratio of a given process taking place at different temperatures, is used to measure the degree of temperature dependence of a given biological process (*Davidson, Janssens & Luo, 2006*; *Mundim et al., 2020*). $Q_{10}$ is an important parameter in predicting the effects of temperature on carbon dioxide release (*Mundim et al., 2020*).

Previous studies on *Helicoverpa* species (eg. *H. armigera* and *H. punctigera*) focused on the effect of temperature on development rate, survivorship and reproduction (*Room, 1983*; *Qayyum & Zalucki, 1987*; *Mironidis & Savopoulou-Soultani, 2008*; *Zalucki & Furlong, 2005*; *Mironidis, 2014*) but not on thermal physiology ($CT_{max}$ and metabolic rate) of different life stages. For example, *H. punctigera* are distinctive in terms of their life stage variation and habitat (*Gregg et al., 2016*, *2019*), just as other insects with stage-based niche changes. Across life stages, an individual will be exposed to a range of temperatures (*Agudelo-Cantero & Navas, 2019*) with different opportunities to escape extreme exposure. In the field, the larvae and adults of *H. punctigera* live on plant surfaces whiles the pupae lives in the soil. Based on this life stage and habitat variations, their tolerance to extreme temperatures may significantly vary among the different life stages. Our study is the first of *H. punctigera* thermal physiology across three life-stages–larvae, pupae and adult.

We assessed the thermal physiology ($CT_{max}$ and metabolic rate) of three different life stages—late instar larvae, pupae, and adult—and sex of *H. punctigera* during temperature ramping using thermolimit respirometry, which employs a flow-through respirometry, with $\acute{V}CO_2$ as a key measure. The technique has an infrared activity detector to monitor moving activity (*Lighton & Turner, 2004*; *Vorhees & Bradley, 2012*). One important advantage of this technique is that $CT_{max}$ can be identified directly from respiratory or activity data. In addition, it allows for $CT_{max}$ to be determined without disturbing the organism (*Vorhees & Bradley, 2012*). The technique, in addition, allows for the measurement of $\acute{V}CO_2$ when exposed to biologically relevant temperature points.

The specific questions we asked are:

(i) what are the differences in $\acute{V}CO_2$ and $CT_{max}$ across different life stages of *H. punctigera* considering the different field conditions each stage experiences? We predict that, larvae and adult of *H. punctigera*, which are exposed to the many microclimatic conditions on host plants surface, would have a higher metabolic rate and $CT_{max}$ than pupae, because, the soil the pupae is burrowed in is more thermally stable than the leaves.

(ii) what are the metabolic rates of the life stages at five different temperatures at 5 °C bins: 25, 30, 35, 40 and 45 °C? Here, we predict that because exposure temperature has an overwhelming effect on the metabolic rate of insects, the metabolic rate among the life stages would be lower at 25, 30 and 35 °C compared with 40 and 45 °C.

(iii) does sex influence the metabolic rate and $CT_{max}$ of adult and pupae of *H. punctigera*? We predict that because of the differences in activity (*e.g.*, host selection and

dispersal in the case of adults) between male and female, sex would influence metabolic rate and $CT_{max}$ of *H. punctigera* pupae and adult.

## MATERIALS AND METHODS

### Insect culture and experimental animals

A laboratory insect culture was established in April 2018 from pupae provided by Tamworth Agricultural Institute, New South Wales Department of Primary Industries. After emergence, the adults were put into cylindrical framed mating cages containing 5% sugar solution in dental wicks held in 50 ml plastic containers. We left female moths to mate and lay fertile eggs for 72 h following protocols outlined by *Gregg et al. (2016)*. Eggs were collected by daily scrolling (by way of folding) of the paper towel which acts as the mating cage wall, followed by cutting the towel containing the eggs into pieces measuring approximately 7cm × 21cm × 11cm. Eggs were cleaned by brief immersion in 0.2% sodium hypochlorite solution for disease prevention, brushed carefully for one to two minutes and then rinsed two times with distilled water onto filter paper placed in a Büchner funnel. The cleaned eggs were put in plastic eggcups coated with artificial diet and kept at 25 °C. Five days after hatching, the larvae were transferred into 35 ml plastic cups ventilated with small holes and containing 10–15 ml of soybean-based artificial diet (*Greene, Leppla & Dickerson, 1976*; *Teakle, 1991*). The insects were reared in the Agronomy insectary maintained at controlled conditions of 70–75% relative humidity, 14 h: 10 h light: dark period and 25 °C, University of New England, NSW, Australia. For a continuous supply of *H. punctigera* individuals at appropriate life-stages, two cultures were maintained at different developmental stages. Before and after each experiment, the larvae, pupae or adult moths used were weighed on an electronic balance (Mettler Toledo XP 404S, Greifensee, Switzerland) with an adjusted accuracy of 0.1 mg. We used fifth instar larvae in our experiment because they were available and relatively easy to manipulate without damaging their cuticle. The average live body mass of the insects used pre-experiment was 189.09 ± 15.58 mg for adults, 348.26 ± 57.57 mg for larvae and 332.38 ± 18.39 mg for pupae. One-week-old pupae were sexed into male and female under a microscope using the genital scars (*Kirkpatrick, 1961*) and used in pupal assays. Finally, virgin adults that were 1 to 2 days post-emergence were used for the experiment: these adults were not fed. Because of the polyphagous nature of the insect, larvae would always have a bite of a meal or would always have eaten in the field, so all measures of metabolic rate include the specific dynamic action (roughly, the cost of digestion) of the larvae. Larvae were fed with the same amount of food and quality ad-libatum, until it was time to use for the experiment.

### Experimental setup

We used standard thermolimit respirometry protocols (*Lighton & Turner, 2004*; *Terblanche et al., 2007*; *Boardman & Terblanche, 2015*; *Andrew, Ghaedi & Groenewald, 2016*; *Ghaedi & Andrew, 2016*). In brief, using a HiBlow HP40 air pump, room air was pushed into sodalime to remove $CO_2$ and Drierite to remove moisture. Air was set to flow at 180 ml min$^{-1}$ using a mass flow controller (Sable MFC-2; Sable Systems, Las Vegas,

NV, USA) and flow control valve (Side-Trak 840L; Sierra Instruments Inc., Monterey, CA, USA) through Bev-a-line tubing to a 30 ml glass cuvette. Carbon dioxide and water in the air that was flowing before and after the glass cuvette was analysed by a two-channel calibrated infrared Li-7,000 analyser (Li-Cor; Lincoln, NE, USA). The scrubbed airstream flowed across the experimental organism in the 30 ml glass cuvette, which was held underwater in a water bath (Grant, GP200-R4), programmed (resolution ± 0.1 °C) with LABWISE software. Cuvettes were placed into two plastic bags to stop any water entering the cuvette. We warmed insects from 25 °C to 55 °C, at 0.25 °C min$^{-1}$ (*Andrew, Ghaedi & Groenewald, 2016*), over 120 min in the 30 ml glass cuvette. The outgoing air from the cuvette then enters the infrared Li-cor analyser again: the $CO_2$ concentration difference of the air before and after it flowed through the cuvette, at one-second intervals is then recorded. A type-T thermocouple attached to a data logger (PicoTech TC-08 data logger with resolution ± 0.025 °C, accuracy ± 0.5 °C) was used to record the temperature in the glass cuvette. The two data recordings (Licor and data logger) were combined using Microsoft excel, aligned using the computer time stamp. An infrared activity detector (AD-1; Sable Systems, Las Vegas, NV, USA) recorded the movement of the test insect (see trace in Supplementary).

## Data Extraction and measurement of $CO_2$ emission rate (metabolic rate) and $CT_{max}$

Using Expedata (Sable Systems International, Las Vegas, NV, USA) version 1.9.2 analysis software, data on $CO_2$ emission rate was extracted and measured from 25 °C to 55 °C (*Boardman & Terblanche, 2015*; *Andrew, Ghaedi & Groenewald, 2016*). We drift-corrected our recordings to baseline recordings from an empty 30 ml glass cuvette made 10 min before and after trial, and converted to ml $CO_2$ h$^{-1}$ (*Andrew, Ghaedi & Groenewald, 2016*; *Ghaedi & Andrew, 2016*). We controlled chamber switching by recording from the cuvette by manually adding/removing the insect from the same cuvette as the baseline. For each replicate insect, V́$CO_2$ (*n* = 10 larvae, 10 pupae, 10 adults) were extracted at these temperature bins (25, 30, 35, 40, 45 °C and $CT_{max}$ ± 0.25 °C). The overall V́$CO_2$ (in ml $CO_2$ h$^{-1}$) was calculated by extracting the area under the curve (integral of $CO_2$ ml h$^{-1}$ *vs* hours) in total over the ramping period from 25 °C to $CT_{max}$ by transforming ppm concentration of $CO_2$ to $CO_2$ fraction and then multiplied by the flow rate. This area was equal to the volume of $CO_2$ produced by each replicate insect in the 30 ml glass cuvette (*Stevens et al., 2010*; *Andrew, Ghaedi & Groenewald, 2016*). The V́$CO_2$ for each replicate insect at each temperature point was extracted and multiplied by 1,000 to give μl $CO_2$ h$^{-1}$ per insect per hour. $CT_{max}$ (*n* = 10 larvae, 10 pupae, 10 adults) was determined by identifying the point at which the spiracular activity ceases *i.e.*, the inflection point in the absolute difference sum (ADS) residuals (*Lighton & Turner, 2004*; *Vorhees & Bradley, 2012*; *Andrew, Ghaedi & Groenewald, 2016*; *Ghaedi & Andrew, 2016*). Because of differences in activity among the life stages, to standardise our analysis, we used V́$CO_2$ data to determine $CT_{max}$ value not activity. Other studies showed that V́$CO_2$ and activities

measurements produced similar estimates of $CT_{max}$ in insects (*Lighton & Turner, 2004*; *Vorhees & Bradley, 2012*). Metabolic rates at 25, 35 and 45 °C for $Q_{10}$, were calculated as the average $\acute{V}CO_2$ over the 5-min period at each temperature *e.g.* start at 25 °C and go to 26.25 °C. The inflection point for activity ADS residuals where motor activity ceases is the activity $CT_{max}$. *Lighton & Turner (2004)* defined ADS as the cumulative sum of the absolute difference between adjacent sampling points.

## Data analysis

We considered our measures of $\acute{V}CO_2$ as a proxy for metabolic rate (*Terblanche, Clusella-Trullas & Chown, 2010*). Statistical analyses were carried out in SAS® 9.4 software (*SAS Institute Inc, 2013*) using the GLIMMIX procedure. $CT_{max}$ and total metabolic rate differences between the life stages were determined using generalised linear models (GLM). Differences in metabolic rate for different temperatures were assessed using generalised linear mixed models (GLMM) with individuals as random factors to balance for the repeated measurement. A one-way analysis of variance (ANOVA) was used to compare the overall $\acute{V}CO_2$ and $CT_{max}$ among the life stages. The $Q_{10}$ (25–35 °C) and $Q_{10}$ (35–45 °C) for the life stages were also recorded similar to the above described procedure. Data were reported as means plus or minus standard error (mean ± SE). In all the models $CT_{max}$ and $\acute{V}CO_2$ of individuals were the response variables with life stage as the explanatory variables. Data was checked for normality and homogeneity of variance. The factors in the model include life stage, sex and body mass as a covariate for metabolic rate. Because sex was not significant for $CT_{max}$ and metabolic rate, data on sex for those parameters were combined in the subsequent analysis. Mass was removed from the final model for $CT_{max}$ since it was not significant. Where there were significant differences, post-hoc analysis was performed using least-square means (LSmeans) and the differences compared, $\alpha = 0.05$ (Tukey-Kramer Adjustment). Mass was log-transformed before analysis. T-test was used to compare treatments under two treatments.

## RESULTS

### Metabolic rate (overall $\acute{V}CO_2$ (ml h$^{-1}$) across life stage of *H. punctigera*

There was a significant effect of life stages on overall $\acute{V}CO_2$ (ANOVA, DF = 2, F = 31.52, $p < 0.0001$). Pupae ($n = 10$) had an overall $\acute{V}CO_2$ of 41% lower than adults ($n = 10$) and 50% lower than larvae ($n = 10$) ($p < 0.0001$, and $p < 0.0001$, respectively; Fig. 1). The $\acute{V}CO_2$ of larvae was 9% higher than adults, but not statistically significant ($p = 0.20$; Fig. 1).

### Upper thermal tolerance ($CT_{max}$) across life stages of *H. punctigera*

There was a significant life stage effect on $CT_{max}$ (ANOVA, DF = 2, F = 43, $p < 0.0001$). Adult ($n = 10$) and pupae ($n = 10$) had a similar $CT_{max}$ ($p = 0.12$; Fig. 2). However, larvae ($n = 10$) had a $CT_{max}$ on average 2.2 °C higher than adults and 1.8 °C higher than pupae ($p < 0.0001$; $p =< 0.0001$, respectively; Fig. 2). Activity $CT_{max}$ and $CO_2$ $CT_{max}$ for adults were similar and not significant; unlike larvae, which had significantly higher $CO_2$ $CT_{max}$ (Table 1).

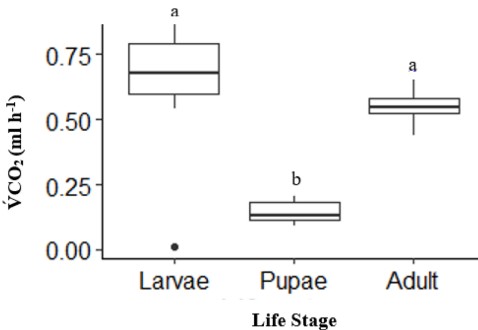

**Figure 1** Box plot showing the Total mean $\dot{V}CO_2$ ml $h^{-1}$ among life stages (larvae, pupae and adult; $n = 10$ for all stages) of *H. punctigera* over complete ramping period of about 120 mins from 25 °C to CTmax following thermolimit respirometry. Box plots with the same letter are not significantly different ($p < 0.05$). The box represents the lower and upper quartiles with the dark line representing the mean in each group. The bars represents the minimum and maximum whiskers, the dot represents an outlier.

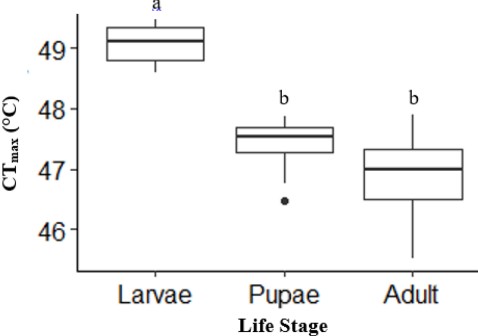

**Figure 2** Box plot showing the Mean $CT_{max}$ identified as the inflection point in the absolute difference sum (ADS) residuals among life stages (larvae, pupae and adult; $n = 10$ for all stages) of *H. punctigera* following thermolimit respirometry. Box plots with the same letter are not significantly different ($p > 0.05$). The box represents the lower and upper quartiles with the dark line representing the mean of the data in each group. The bars represents the minimum and maximum whiskers, the dot represents an outlier.

**Table 1** Anova summary information for Activity $CT_{max}$, $CO_2$ $CT_{max}$ and mean $CT_{max}$ between two life stages (Larvae and Adults) of *H. punctigera* following thermolimit respirometry.

| Stage | DF | Sum Sq. | Mean Sq. | F | p | Mean $CT_{max}$ Life stage | |
|---|---|---|---|---|---|---|---|
| | | | | | | Larvae | Adults |
| Activity $CT_{max}$ | 1 | 8.3 | 8.28 | 0.21 | 0.65 | 44.1 ± 1.3 | 43.1 ± 2.5 |
| $CO_2$ $CT_{max}$ | 1 | 133.1 | 133.08 | 4.27 | 0.05 | 49.0 ± 0.3 | 43.5 ± 2.4 |

## $\dot{V}CO_2$ of *H. punctigera* life stages at different temperature points

Temperature effect on mean $\dot{V}CO_2$ was high in adults and larvae compared to pupae (Table 2) at the different temperature points. The mean $\dot{V}CO_2$ of pupae was significantly

**Table 2 Mean ± s.e. V́CO$_2$ (µl CO$_2$ h$^{-1}$) over different temperature points among life stages (larvae, pupae and adult).**

| Life Stage | Mean V́CO$_2$ (µl CO$_2$ h$^{-1}$) | | | | | | | | | | | |
|---|---|---|---|---|---|---|---|---|---|---|---|---|
| | 25 °C | 95% CL | 30 °C | 95% CL | 35 °C | 95% CL | 40 °C | 95% CL | 45 °C | 95% CL | CT$_{max}$ | 95% CL |
| Larvae | **0.40 ± 0.04a** | [0.31–0.50] | **0.44 ± 0.04a** | [0.43–0.62] | **0.53 ± 0.04a** | [0.57–0.76] | **0.70 ± 0.06a** | [0.54–0.81] | **1.03 ± 0.22a** | [0.83–1.78] | **1.03 ± 0.14a** | [0.90–1.50] |
| Pupae | **0.08±0.03b** | [−0.03 to 0.11] | **0.11 ± 0.03b** | [0.02–0.15] | **0.14 ± 0.03b** | [0.08–0.22] | **0.19 ± 0.04b** | [−0.01 to 0.16] | **0.24 ± 0.16b** | [−0.09 to 0.60] | **0.23 ± 0.10b** | [−0.05 to 0.37] |
| Adult | **0.21±0.04b** | [0.16–0.36] | **0.31 ± 0.04c** | [0.21–0.41] | **0.37 ± 0.04c** | [0.22–0.43] | **0.71 ± 0.06a** | [0.71–1.01] | **1.47 ± 0.24c** | [0.91–1.92] | **1.23 ± 0.15a** | [1.01–1.65] |

Note:
Values in bold indicate significant differences. Letters indicate significant differences among life stages. The 95% CL is 95% confidence limit.

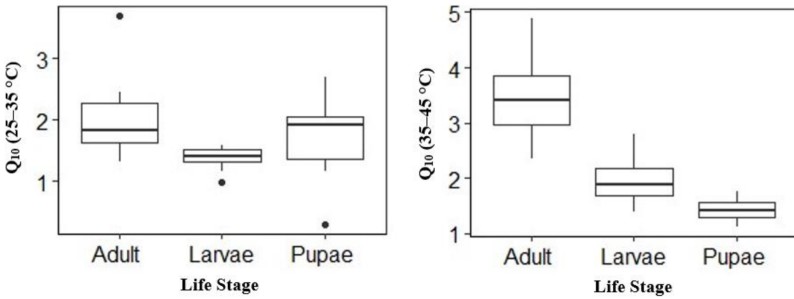

**Figure 3 Box plots comparisons of *H. punctigera* life stages (adult, larvae and pupae; *n* = 10 for all stages) Q$_{10}$ (25–35 °C) and Q$_{10}$ (35–45 °C).** The box represents the lower and upper quartiles with the dark line representing the mean. The bars represents the minimum and maximum whiskers, the dots represent outliers.

different from that of adult and larvae at all temperature points except at 25 °C (Table 2). The mean V́CO$_2$ of each of the life stages doubled at both 40 °C and 45 °C (Table 2) before CT$_{max}$. Although the mean V́CO$_2$ between adults and larvae was not significantly different at 40 °C, adults had a 0.01 µl h$^{-1}$ higher mean V́CO$_2$ than larvae (Table 2). At equilibrium (25 °C) and 30–35 °C larvae had 0.19 µl h$^{-1}$, 0.13 µl h$^{-1}$ and 0.16 µl h$^{-1}$ higher than adult (Table 2) respectively. The Q$_{10}$ 25–35 °C values for larvae and adults were lower compared to Q$_{10}$ 35–45 °C–28% lower for larvae (Q$_{10}$: 25–35 °C 1.40 ± 0.06 to Q$_{10}$ 35–45 °C 1.95 ± 0.14), 41% lower for adults (2.01 ± 0.22 to 3.42 ± 0.24) but 32% higher for pupae (1.74 ± 0.21 to 1.42 ± 0.98) (Fig. 3). Among life stages the Q$_{10}$: 25–35 °C for adults was significantly higher (*p* = 0.04) compared with larvae, but not pupae (*p* = 0.53; Fig. 3). The Q$_{10}$: 25–35 °C was not significant between larvae and pupae (*p* = 0.35; Fig. 3). The Q$_{10}$: 35–45 °C values were not statistically different among larvae and pupae (*p* = 0.07; Fig. 3) but significant among larvae and adults as well as pupae and adults (*p* = 0.00, *p* = 0.00; Fig. 3) respectively.

## The influence of sex on upper thermal limit (CT$_{max}$) and metabolic rates V́CO$_2$ (ml h$^{-1}$) of adult moth and pupae

There was no sex effect (ANOVA, DF = 3, F = 0.15, *p* = 0.92; Fig. 4) on the CT$_{max}$ of adults (males: 46.92 °C ± 0.29 °C and females: 46.72 °C ± 0.29 °C) and pupae (males: 47.47 °C ± 0.29 °C and females: 47.32 °C ± 0.29 °C). For metabolic rate assessments, there was no significant difference between female and male adults as well as between female and male pupae (t = −0.57, *p* = 0.93; Fig. 5) and (t = 1.38, *p* = 0.52; Fig. 5) respectively.

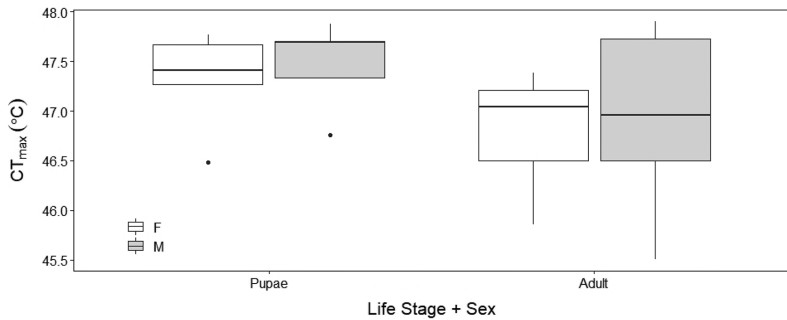

**Figure 4 Box plots of upper thermal limit (CT$_{max}$) identified as the inflection point in the absolute difference sum (ADS) residuals among Life Stage (pupae and adult) and sex (male–M and female–F) of *H. punctigera* following thermolimit respirom.** The box represents the lower and upper quartiles with the dark line representing the mean. The bars represent the minimum and maximum whiskers, the dots represent outliers.

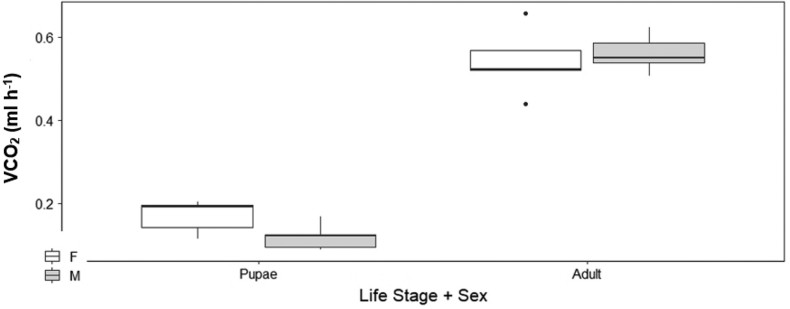

**Figure 5 Box plots of V́CO$_2$ ml h$^{-1}$ among Life Stage (pupae and adult) and sex (male–M and female–F) of *H. punctigera* over the complete temperature ramping period of about 120 mins from 25 °C to CT$_{max}$ following thermo.** The box represents the lower and upper quartiles with the dark line representing the median. The bars represents the minimum and maximum whiskers, the dots represent outliers.

## DISCUSSION

Here, we measured the upper thermal tolerance limits (CT$_{max}$) and metabolic rates at different temperature exposures across three life stages larvae, pupae, and adults and between sexes of *H. punctigera* using thermolimit respirometry. These life stages experience different temperatures because of the different habitats they naturally occupy, enabling an ecologically relevant investigation of their metabolic rates at different temperatures. We found that CT$_{max}$ differed among the life stages of the insect with larvae exhibiting significantly higher CT$_{max}$ compared with pupae and adults (Fig. 2). Similar to the findings presented here, studies by other authors have shown that CT$_{max}$ differs among life stages of insects including *Paractora dreuxi*, *Sirex noctilio* and *Tenebrio molitor* (*Marais, Terblanche & Chown, 2009*; *Vorhees & Bradley, 2012*; *Li et al., 2019*). The higher CT$_{max}$ of the larvae compared with adults however, contradicts the findings of *Vorhees & Bradley (2012)* and *Li et al. (2019)* in *T. molitor* and *S. noctilio* respectively, where adults had a higher CT$_{max}$ than larvae, although, the insects are of different taxa. The differences in the CT$_{max}$ between the life stages suggest differences in upper
thermal tolerance in this species with larvae having the highest thermal tolerance to high temperature (*Marais, Terblanche & Chown, 2009*).

The $CT_{max}$ differences among the life stages could be caused by several different factors such as interactions of the microclimatic conditions experienced by the life stages and the level to which they compensate their physiology to avoid harsh or extreme temperatures (*Vorhees & Bradley, 2012*). However, this would not be the case here, as all insects were kept at a constant 25 °C, so a more evolutionarily relevant mechanism must be in train here. The results imply that, larvae, pupae and adults have the ability in the wild to adapt and cope differently.

The significantly higher $CT_{max}$ exhibited by larvae is not surprising because larvae in their natural environment are generally more exposed to extreme/hot temperatures, just above the soil surface and while feeding on plants, and, less able to behaviourally adapt to extreme periods. Hence the need to modify their physiological response to such extreme periods of heat conditions for survival (*Andrew et al., 2013a*, *2013b*; *Andrew, Ghaedi & Groenewald, 2016*). Even though the insects used in the study were laboratory-reared, they were most likely to exhibit similar characteristics like those in the field, because there is little genetic variability among *H. punctigera* population (*Daly & Gregg, 1985*). The limitation of our study is that; the low genetic diversity in the wild in the insect does not imply that laboratory populations will not be adapted after many generations. We acknowledge that the climatic variability experienced in the wild could further influence in these thermal tolerance traits, but is unlikely to affect the clear life-stage differences.

The decline in the upper thermal limit between larvae, pupae and adults in that order (Fig. 2) suggests that the insects upper thermal limit could be influenced by ontogeny as observed in *Drosophila melanogaster* (*Bowler & Terblanche, 2008*), although we did not test for age effects in a single life stage. This upper thermal limit decline across the life stages however, does not always follow a logical order as in some insect's *e.g. Drosophila buzzatii*; heat tolerance was high in pupae followed by eggs and larvae (as cited in *Vorhees & Bradley, 2012*). Therefore, caution must be made against the generalisation concerning the effects of ontogeny on heat resistance (*Vorhees & Bradley, 2012*), although we should expect ontogenetic variation (*Bowler & Terblanche, 2008*). One would have expected in part, no significant difference in the $CT_{max}$ between larvae and adults because of the several factors like the microclimate conditions they experienced and the extent to which they can use their physiology to avoid the extreme temperatures. The lower $CT_{max}$ of the adults compared with the larvae may be because the adults have a higher ability to avoid extreme temperatures by flying away from areas of extreme temperatures compared to larvae. The results suggest that there are ecological reasons for different thermal preferences: larvae are more sedentary so it is harder for them to move away from hotter temperatures they may be exposed to; adults are more mobile, so can move to preferred temperatures refuges in the evening so have a lower thermal threshold. Because of the ectothermic and regional heterothermic nature of insects, high temperatures could speed up growth and development of insects (*Andrew, Hart & Terblanche, 2011*; *DeVries, Kells & Appel, 2016*). The higher $CT_{max}$ of the larvae could assist them to grow

faster in warmer conditions like summer if food, and most importantly water, are not limiting factors.

The $CT_{max}$ exhibited by the life stages is significantly greater than the highest recorded maximum daily mean ambient temperature in Tamworth region where the population originated from (40.9 °C) (NicheMapR). However, the microclimatic conditions that *H. punctigera* could be exposed to may be much higher than this, particularly during the middle of the day (*Andrew et al., 2013a*), and can put the insects under physiological stress that can change their interactions under changing climate (*Dawson et al., 2011*). The higher $CT_{max}$ exhibited by the insect could be the reason for which it is able to occupy a wide range of climatic conditions including inland of Australia, and spread in all states of Australia (*Gregg et al., 2016, 2019*).

Our results show that upper thermal limits were not significantly different between males and females. The similar thermal limits demonstrated by males and females indicates that sex has minimal influence on $CT_{max}$ (Fig. 4), similar to the findings in bumble bees (*Bombus terrestris*) (*Oyen, Giri & Dillon, 2016*); suggesting that *H. punctigera* sex ratios will not change with adult exposure to climatic extremes.

Insects as other animals require energy to forage, survive, grow and reproduce (*Terblanche, Klok & Chown, 2004*; *White et al., 2019*). The processes that cause variation in metabolic rate apart from ambient temperature, body mass is also an important factor that contributes to metabolic rate of insects (*White et al., 2019*). Insects show a range of thermoregulatory capacities: ectothermic, regional heterothermic, poikilothermic and endothermic (*Heinrich, 1993*; *Gullan & Cranston, 2014*; *O'Neill, Kemp & Johnson, 1990*) their activity and physiological processes are primarily influenced by ambient temperature (*Nguyen et al., 2014*). Our results revealed $\acute{V}CO_2$ varied among the life stages at the different temperature points. The differences in the mean $\acute{V}CO_2$ among the life stages suggests that the metabolic demands of the life stages differ. Several factors may account for the differences in the metabolic rates among the life stages; including ontogeny, upper thermal tolerance, activity, body mass and feeding (*Terblanche, Klok & Chown, 2005*).

Fat body tissues play crucial roles in the life of insects, as they are involved in multiple metabolic functions like energy storage and utilisation in reply to the energy call or request of the insect (*Arrese & Soulages, 2010*). Muscle tissue usually has a higher metabolic rate than fat body tissue. Muscle tissues are active and burns calories even at rest unlike fat body for storing excess energy.

The significantly higher $\acute{V}CO_2$ of larvae and adult compared with pupae could be due to low amount of fat body tissues in the larvae and high muscle tissues in the case of adult. Another reason for the high metabolic rate may be due to the high activities of larvae and adult compared with pupae, similar to that reported by *DeVries, Kells & Appel (2013)* for the bed bug, *Cimex lectularius*. Size also affects the metabolic rate of an organism (*Terblanche, Klok & Chown, 2004*; *Quinlan & Gibbs, 2006*; *DeVries, Kells & Appel, 2013*), and was similar in our case, as size was significant (Figure 3 in Supplemental Material). Pupae, which has a higher $CT_{max}$ than adults, had a comparative lower $\acute{V}CO_2$. The lower $\acute{V}CO_2$ may be an indication of lower stress at the higher temperatures and or maybe due to more fat tissues in pupae. It is plausible that it may be an adaption

strategy for survival to dry conditions (*Terblanche, Klok & Chown, 2004*; *Quinlan & Gibbs, 2006*). The high amount of fat tissues in the pupae because of lipid reserves needed to provide energy during the extended non-feeding periods could be the reason for the low metabolism (*Arrese & Soulages, 2010*). Because pupae do not have a source of free water underground, they must conserve water by reducing respiration (Table 1 in Supplemental Material) (*Harrison, 2009*).

As the effect of temperature on metabolic rates of insects is widely reported (*Terblanche, Klok & Chown, 2004*; *Terblanche & Chown, 2007*; *Hill et al., 2016*), only a few studies looked at that at different temperature points for different life stages for *H. punctigera*. Our results on metabolic rate between life stages considering the different temperatures (Table 2) and $Q_{10}$ (Fig. 3) indicates that there was a correlation between temperature rise and $\acute{V}CO_2$ release on the life stages during the thermolimit respiratory (Table 2). The $Q_{10}$ values for all the life stages except pupae almost doubled from $Q_{10}$: 25–35 °C to $Q_{10}$: 35–45 °C, with adults having the highest thermal sensitivity. This increase in metabolic rate as temperature increases is consistent with other studies (*Terblanche et al., 2005*; *Terblanche & Chown, 2007*; *Andrew, Ghaedi & Groenewald, 2016*). However, the negative effects associated with increasing temperatures on metabolic efficiency could be reduced on adults, as they are able to compensate for short-term extreme microclimate temperatures by flying away. Although, increase in temperatures result in high metabolic rate, the total energy assigned for certain actions, as specific dynamic actions are not dependent on temperature (*McCue et al., 2016*; *Wang et al., 2002*). It is possible that as insects enlarge their geographic radius or scale, they will face abiotic and biotic stress.

For better comparison among life stages, we measured $\acute{V}CO_2$ at $CT_{max}$, and in all cases, $\acute{V}CO_2$ among life stages was higher. Even though the $\acute{V}CO_2$ at $CT_{max}$ increased in all the life stages, adults $\acute{V}CO_2$ were higher than larvae and pupae. *Harrison (2009)* reported that tracheated insects have the ability to carry or move gases by opening their spiracles. Short periods of spiracles closure follow a continuous carbon dioxide exchange (*Harrison, 2009*). The higher metabolic rates due to higher stress levels may be due to a decrease between spiracle opening and closing. Therefore, the higher $\acute{V}CO_2$ of adults at $CT_{max}$ may be due to their high activity rate and a decrease between its spiracle opening and closing compared to larvae and pupae (*Harrison, 2009*; *Boardman & Terblanche, 2015*). The lower metabolic rate of pupae at $CT_{max}$ compared to larvae and adults could not be explained beyond a physiological adaption strategy to reduce physiological cost by reducing respiration and conserve water at this stage (*Harrison, 2009*), similar to what *Merkey et al. (2011)* reported in *Drosophila melanogaster* pupae.

The lower $\acute{V}CO_2$ of adults compared to larvae at 25 °C could suggest that adults have a lower resting metabolic rate than larvae at that point. The higher resting metabolic rate of larvae could be the result of modification of the energy allocation between competing functions such as maintenance for continuous development. Additionally, the lower $\acute{V}CO_2$ of adults may be as result of the occurrence of discontinuous gas exchange in the adults at 25 °C. Discontinuous gas exchange is associated with low metabolic rates. In contrast, age, as proven to affect metabolic rates in some insects like *Glossina pallidipes* (*Terblanche, Klok & Chown, 2004*) did not apply in our case, as larvae had a lower

$\acute{V}CO_2$. The high $\acute{V}CO_2$ produced by the larvae could imply a high-energy utilisation (*Terblanche, Klok & Chown, 2004*) for development. This high-energy utilisation can provide a deep understanding into the life histories of the life stage across a variety of environments. Therefore, this high-energy utilisation may translate to an increase in food consumption and more considerable plant damage (*Deutsch et al., 2018*). The metabolic rate of pupae and adults was not influenced by sex, suggesting that the cost of living of the dispersing adults and pupae of both sexes are similar. Similar work by *Terblanche, Klok & Chown (2004)* found out that metabolic rate did not differ between males and females of *Glossina pallidipes* and *G. morsitans*.

## CONCLUSION

The present study provides vital information on how the different life stages of *H. punctigera* will physiologically perform in a changing thermal environment. It also provides data for predictions about metabolic rates of *H. punctigera* among life stages. The study revealed that $CT_{max}$ of larvae was significantly higher than those of pupae and adults, and the metabolic rate of pupae was significantly different from those of larvae and adults. Also, sex did not influence $CT_{max}$ and metabolic rate. The results suggest that the performance and survival of the different life stages of *H. punctigera* could differ. This indicates that life-stage needs to be taken into account when assessing responses to a rapidly changing change, as ontogeny influences vulnerability and exposure. This finding can be used to predict the potential spread of the insect to other new environments and can indirectly help in management planning.

## ACKNOWLEDGEMENTS

Sarah Hill assisted with the setup, calibration, maintenance and protocols of the thermolimit respirometry system. Zac Hemmings assisted in the data extraction, analysis and initial interpretations of the raw-data outputs. We thank Bianca Boss-Bishop, Sarah Hill, James O'Hanlon, Zac Hemmings, Alva Rebecka Curtsdotter and Behnaz Ghaedi for their comments on earlier versions of the manuscript. We also thank NSW DPI for providing the insects for the culture.

### Funding

University of New England (UNE) provided funds as scholarship to Samuel Abukari Bawa as well as for equipment and consumables used in the study. The funders had no role in study design, data collection and analysis, decision to publish, or preparation of the manuscript.

### Grant Disclosures

The following grant information was disclosed by the authors:
University of New England (UNE).

## Competing Interests

Nigel Andrew is an Academic Editor of PeerJ.

## Author Contributions

- Samuel A. Bawa conceived and designed the experiments, performed the experiments, analyzed the data, prepared figures and/or tables, authored or reviewed drafts of the paper, and approved the final draft.
- Peter C. Gregg conceived and designed the experiments, authored or reviewed drafts of the paper, and approved the final draft.
- Alice P. Del Soccoro conceived and designed the experiments, authored or reviewed drafts of the paper, and approved the final draft.
- Cara Miller conceived and designed the experiments, analyzed the data, authored or reviewed drafts of the paper, and approved the final draft.
- Nigel R. Andrew conceived and designed the experiments, authored or reviewed drafts of the paper, and approved the final draft.

## Data Availability

The raw data is available in the Supplemental File.

## Supplemental Information

Supplemental information for this article can be found online at http://dx.doi.org/10.7717/peerj.12479#supplemental-information.

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
