# Peer review of "Estimating the differences in critical thermal maximum and metabolic rate of Helicoverpa punctigera (Wallengren) (Lepidoptera: Noctuidae) across life stages"

_PeerJ, doi:10.7717/peerj.12479_

## Round 0.1 · original submission · Major Revisions

Your manuscript was reviewed by three experts in the field and although all three indicate it is an interesting study, they all raise points that must be clarified. Overall, please carefully read over the manuscript for spelling, punctuation, grammar, and conciseness. In the introduction and discussion, please include other literature relevant to the topic. Please comment on/justify the low sample sizes used in the experiments, and clarify the issues raised with the raw data provided. I look forward to receiving your revisions and your point-by-point rebuttal.

Reviewer 1 ·

Basic reporting

Generally the writing is clear, and the methods were particularly well-written, and the authors have done well to provide references to support most of their arguments. I have three concerns:

1. Please include metadata with your raw data that explains each column title, and each factor. For example, there is an ambiguous column titled 'Log transform' and none of the columns describe the units of measure. Also the precision of their raw data is not consistent across columns and rows (some V-dotCO2 data is reported to X.XXXX, while others are reported to X.XXXXXXXX). I also think an example trace showing the raw VCO2 data and activity within an example recording for one insect, as well as the inflection point they used to determine the CTmax is important to include.

2. In places, the writing in the introduction and discussion can be unclear, and sometimes difficult to link to the overall study design. The arguments/language can come across as grandiose when describing the gaps in knowledge that exist in the literature, as well as the potential applications of CTmax and metabolic rate data It's also not always clear why some arguments are being made (e.g. about gas exchange patterns in insects) in the introduction when this information isn't directly address by this study. The links between CTmax and predicting an insect's vulnerability to climate change are contentious, and not always well-supported. I encourage the authors to do more to qualify these connections in the introduction.

Overall the authors write well, follow structural conventions and provide their raw data; this manuscript was often and enjoyable read. However, I believe the above concerns should be addressed before this work published. I have also provided specific suggestions/comments by line number below.

Experimental design

In general, I think the authors' methods are well-described and provide sufficient detail such that the work could be replicated. Their research questions could be better stated as well-justified hypotheses, and this would help the reader understand relevance and meaning of their work. I do have two major concerns:

1. The authors' research questions are OK, but a meaningful justification for their predictions is missing. Two of their predictions describe 'phenotypic plasticity' as an explanation for the phenomena they will observe, yet their insects were reared under a single constant temperature. The authors are measuring immediate responses to temperature, without an opportunity for phenotypic plasticity to occur, so it's not clear to me how their experiment can address these predictions. I'm also not convinced that the explanations for their predictions are the most apparent. For example, they predict that "because insects are plastic in their responses to temperature, the metabolic rate among the life stages would be lower at the ecologically relevant condition compared with the stress conditions" (Lines 161-163). However, their ‘stress conditions’ are high temperature and their ‘ecologically relevant conditions’ are lower temperatures. Temperature has an overwhelming effect on the metabolic rate of an ectotherm due to thermodynamics, so of course metabolic rate will almost certainly increase, but this not primarily because of plasticity or stress per se.

2. While their respirometry methods are generally well-written and explained, I have concerns with how the carbon dioxide production data are extracted/described: At line 259 and in their figures, the authors describe metabolic rate as (V-dotCO2 ml h-1) which is a volumetric rate (i.e. V-dot). This is a standard proxy for metabolic rate, and in my experience, V-dotCO2 is usually calculated from the mean of the CO2 ml h-1 trace of flow-through respirometry data. However, at lines 229-230 they describe calculating (V-dotCO2) by extracting the *integral* of the volumetric rate. But the integral of a volumetric rate is a volume (ml). It’s not clear to me how they move from this volume (ml) to the volumetric rates (ml h-1) reported in your figures and results. Are they dividing the volume of CO2 produced by the time over which the integral was taken? The authors need to do more here to explain this calculation/data extraction as their data extraction seems non-standard for flow-through respirometry.

This is made more confusing by the fact that the authors' estimates of overall VCO2 (Figure 1, Figure 5) are much lower than those reported for their temperature-VCO2 curves (Figure 3, Table 1)? This is confusing: the authors need to clarify how these two sets of measures of VCO2 were calculated, if they intend for them both to be proxies of metabolic rate, and clarify the discrepancy in their values.

The authors also account for the effect of body mass (which differs between life stages) on metabolic rate by including it as a covariate but never describe if this is a significant covariate in the results (albeit, they do mention this briefly in the discussion).

It’s also not clear what gas exchange patterns the authors' insects use (discontinuous, continuous, etc. until a brief not in the discussion) which can have important consequences for how VCO2 data should be appropriately extracted.

Overall I believe the authors data have been collected in a technically rigorous manner, and their methods mostly well described. However, I would like to see the above concerns addressed before this work published. I have also provided specific suggestions/comments by line number below.

Validity of the findings

The authors provide the underlying data, although, I believe an example trace of VCO2, the ADS, activity, etc for each life stage would be useful to illustrate their approach to data extraction to the reader.

At times, I'm not entirely convinced of the explanations the author's offer for the patterns in their data (e.g. amount of fat body explaining variation in metabolic rate between life stages) but the authors do a good job of using hedges when they are speculating.

Additional comments

Below I provide specific comments and suggestion, by section and line number, which I believe will improve your manuscript:

Abstract

I think you could revisit the abstract. As it stands, your abstract doesn’t make clear to me why you measured VCO2 other than to estimate CTmax using thermolimit respirometry. You report differences between the mean CO2 emission rate of each life stage in the abstract, but it’s not clear how or why this information is useful to the reader.

• Line 17: “As many insects…” should probably be “As most insects…”
• Line 19-20: “…could elicit different physiological and behavioural responses…” This is a bit too general, I think. What physiological and behavioural responses are you interested in assessing in this manuscript? Do you even assess any behavioural responses here? Suggest revisiting this section of the abstract.
• Line 21 and throughout! : “(V́CO2)” Please use a proper v-dot here, and throughout the manuscript. I appreciate that this may happen at the proofing stage.

Introduction

• Line 58: “Temperature ramping rate can…” Suggest pluralize ‘rate’.
• Line 84-85: “Depending on the microclimatic exposure, the physiology of insects can modify” Strange, overly passive sentence. Suggest “The physiology of insects can be modified by the microclimate they experience” or similar.
• Line 85-87: “Yet, the right predictions about the future are narrow by our limited understanding of eco‐physiological mechanisms and their interactions with evolutionary processes”. Awkward wording, also not clear what predictions (about what?) you are referring to here. Suggest “Predicting XXX in changing climates is made difficult by our limited upstanding of the of eco‐physiological mechanisms and evolutionary processes underlying thermal phenotypic plasticity” or similar?
• Line 93-94: “The adult can travel a long distance and can adapt to different environmental conditions including inland Australia”. Please clarify and describe this adaptation? Is this local adaptation, or are you referring to plasticity here?
• Lines 88-102: While I appreciate some species life-history details, this paragraph can be shortened, and re-arranged to emphasis the details provided in its last two sentences. Your reader will be interested in the thermal biology of these species, along with some basic details of its life cycle, but probably not development times, number of instars, or event its changing diet over development.
• Line 103-110: This paragraph is verbose, and even grandiose in places, suggest you need shorter, declarative statements. You should also align this section with the aims of your work – how are you planning to use your estimates of metabolic rate? I would keep this in the context of why eco-physiologists measure temperature-metabolic rate curves (and less generic statements about cost of living, energy theory, etc.). In places the grammar needs to be corrected and clarity improved (e.g. “…with numerous variables that can be pull out and compared” might be better stated as “Metabolic rate is a complex trait with several sources of variation such as X, Y, and Z” or similar. Also parts of this section are paraphrased in your discussion.
• Line 111: “Metabolic rate can respond quickly to changes in ambient temperature, activity and body mass;” I agree with you about temperature and activity, but I don’t think saying the metabolic rate “responds quickly to changes in body mass” is accurate. While metabolic rate does indeed depend on the size of the animal, growth is generally slow relative to changes associated with temperature and activity.
• Line 123-127: Why are you describing gas exchange patterns when you don’t discuss discontinuous, or continuous gas exchange in this manuscript? I appreciate that there are events that happen in gas exchange during thermolimit respirometry in response to temperature, but this isn’t really the same as discontinuous or continuous gas exchange patterns, which are primary about the opening and closing of spiracles? Also you don't describe the gas exchange pattern of your insects until discussing adults in the discussion - what was the pattern of larvae and pupae? How did you account for gas exchange patterns when extracting VCO2 data?
• Line 134-137: It’s not immediately clear why knowing the metabolic rate of different life stages is helpful in the context of applied pest management. I think you can do more to link these two concepts here.
• Line 141-142: ‘across key life-stages’ What makes a life stage key? I would argue they are all key to the insect. Suggest just changing this to “across three life-stages”.
• Line 148-149: “…it also sets the temperature and exposure time according to the body mass and surface area of the insect” Not clear to me what you mean by this. I don’t believe thermolimit respirometry specificaly adjusts the temperature or exposure time in relation to the surface area of the insect? Also, be careful about how you describe thermolimit respirometry in places, for example, I would argue that the advantage of the fact that “ CTmax can be identified directly from respiratory and or activity data” is that is in an objective estimate.
• Line 157: “We predict that because CTmax is highly plastic, larvae and adult of H. punctigera, which are exposed to the many microclimatic conditions on host plants surface, would have a higher metabolic rate and CTmax than pupae” Not clear to me how you reach this prediction – aren’t your animals reared under laboratory conditions, so not exposed to these variables (and thus no opportunity for phenotypic plasticity)? You even seem to acknowledge this in your discussion (Lines 316-317). I would also argue that the fact pupae have lower metabolic rates has less to do with microclimate or plasticity, and more to do with the physiology of metamorphosis.
• Line 161-163: “we predict that because insects are plastic in their responses to temperature, the metabolic rate among the life stages would be lower at the ecologically relevant condition compared with the stress conditions.” It isn’t clear to me how this prediction is relevant to your study. You aren’t assessing plasticity at all, but measuring immediate responses to temperature of animals reared under identical conditions in the laboratory. Your ‘stress conditions’ are high temperature and your ‘ecologically relevant conditions’ are lower temperatures. Temperature has an overwhelming effect on the metabolic rate of an ectotherm due to thermodynamics, so of course metabolic rate will almost certainly increase regardless of plasticity or stress. I think you need to revisit these predictions to make sure that you can address them in the context of your study.
• Line 164-165: “because of the differences in activity or physiology between male and female” What differences? Why? Suggest doing more to justify this prediction, or at least, describing these differences in your paragraph about the species life history.

Materials and methods

• Line 175: “Moths were left for 72 hours to allow for mating, and then for fertile eggs to be laid.” It’s not clear to me if this is parallel structure in your writing (i.e. you are describing two periods of 72 hours, a single period of 72 h with both mating and egg laying, or a period of 72 h followed by some indefinite period of egg laying after this 72 hours). This is not a pressing matter in the methods, but I suggest “We left female moths to mate and lay for 72 hours…” or similar might be more clear.
• Lines 180-181: “…artificial diet.” You introduce artificial diet at this line, and again at line 182, but don’t describe that it is a soybean diet or provide a reference until line 183. Suggest moving up the soybean description and references to the first mention of diet at lines 180-181.
• Line 183-184: “…UNE…” suggest using the full name and country (i.e. University of New England in Australia) at first mention for an international audience that may not necessarily be familiar with the institutions in Australia!
• Line 188: “….(Mettler Toledo XP 404S, made in Switzerland)…” Suggest removing ‘made in’ and providing the details of location (e.g. Town of XXX, Switerland). Also, missing a space between value and units on this line (i.e. ‘0.1mg’ should be ‘0.1 mg’).
• Line 189: “…larval assay…” This wording is a bit awkward. Suggest this might be better stated as “We used fifth instar larvae in our experiments because they were available” or similar.
• Line 189: “Average live body mass….” Suggest adding an article at the start of this sentence (i.e. “The average live body mass…”).
• Line 195: “…so for ecologically relevant purposes, the larvae were fed…” This caveat is probably fine, although it may be a bugbear for some respirometry users/readers. I think the most important thing is that you: (1) mention the caveat (which you’ve done); and (2) make clear that all measures of metabolic rate are confounded by the specific dynamic action (roughly, the cost of digestion) of the larvae. So you aren’t really measuring resting metabolic rate, as it is typically defined, and need to be clear about that. While pupae generally have a lower metabolic rate than other life stages, this difference will be even greater if larvae are feeding. You'll need to be very careful about any comparisons to measures from your larvae.
• Line 198: “Thermolimit respirometry protocols followed standard protocols”. Suggest “We used standard thermolimit respirometry protocols” or similar to avoid awkward repeating here.
• Line 200: “…pushed into two scrubber columns:”. Suggest making clear these are in sequence.
• Line 209: “The temperature was ramping…” The tense is a bit awkward here. Suggest, “We warmed insects from 25 degC to 55 degC at 0.25 degC min-1 over 140 minutes” or similar.
• Line 222: “…drift-corrected for baselines…” This is the first mention of baseline recordings. I suggest explaining these a bit (e.g. “We drift-corrected our recordings to baseline recordings from an empty cuvette made XX min before and after each trail” or similar). How did you control chamber switching? Based on past work from your group I think you are recording from the empty cuvette by manually adding/removing the insect from the same cuvette as the baseline, rather than a multiplexor or similar? Including a sentence like the one I describe above could improve clarity.
• Line 222 and 217: “Test animal…”, “…test organism…” (and elsewhere) maybe just “…insect…” works well in this cases?
• Line 225 “…was measured at these three temperature points…” Was this the same recordings as the thermolimit respirometry? I believe it is based on context, but I think you should watch the ambiguous use of the word ‘measured’ here. Suggest “V-dotCO2 data were extracted at three temperatures…” or similar
• Line 226-227: Suggest these lines could benefit with some reference to microclimate conditions for the region?
• Line 236: I appreciate that it’s impossible to detect the inflection point of ADS-residuals channel for immobile pupae. However, I’m curious how similar your estimates of CTmax are between VCO2-ADS and the activity-ADS approaches for your mobile larvae and adults. Please consider including these data for larvae and adults, even if only in the supporting material.
• Line 244: You are now referring to metabolic rate rather than V-dotCO2 in your writing. I think you need to make a brief declarative statement (e.g. “We considered our measures of VCO2 as a proxy for metabolic rate…” or similar) at the start of your ‘Data analysis’ section to connect these two concepts.


Results and Figures

• Line 258 and elsewhere, including figure axes labels: “…(ml h-1)…” You need to subscript ‘-1’ in ml h-1 (i.e. ‘ml h-1’) here and throughout this ms.
• Line 259 and through results: “…(F=225.17)…” Suggest including degrees of freedom with your F and t statistics.
• Line 259 and Figure 1: ‘…overall VCO2…’ This is from the overall integral the CO2 trace for the entire thermolimit respirometry recording, correct? I’m not sure if this is the best way to compare the metabolic rate between life stages of an insect, particularly because this measure will be dependent on the Q10 (thermal sensitivity) of metabolic rate. Why not only analyze the temperature-metabolic rate curves (e.g. Figure 3) instead, as you do starting at line 270?
• Line 260: “84% lower than adults…and 86% lower than larvae…” Please double-check these percentages as they are very similar. Figure 1 and line 261 (describing a 23% difference between adults and larvae) leads me to believe that the difference in metabolic rate between larvae and pupae should be greater than the difference in metabolic rate between adults and pupae.
• Figure 1: Consider rearranging the order of your x-axis such that the life stages appear in order of time (i.e. Larvae, Pupae, Adults) rather than alphabetically. I appreciate that R (and some other statistical software) orders factors alphabetically by default but this is usual easy enough to reorder. In the y-axis label, V-dotCO2 needs the dot, ‘2’ needs to be subscripted, and ‘-1’ needs to be superscripted. Correct figure caption “over complete ramping period” to “over the compete ramping period”, and “box plot with” to “box plots with”.
• Figure 2: “…in the absolute difference sum (ADS) residuals…” needs to identify what the ADS residuals are of (i.e. V-dotCO2). Correct figure caption “box plot with” to “box plots with”. Consider rearranging the order of your x-axis such that the life stages appear in order of time (i.e. Larvae, Pupae, Adults) rather than alphabetically.
• Figure 3: ditto several of my comments for Figures 1 and 2.
• Figure 4 and Figure 5: In the y-axis label, V-dotCO2 needs the dot, ‘2’ needs to be subscripted, and ‘-1’ needs to be superscripted.
• Line 271: “…temperature effect on mean VCO2 was high in adults and larvae compared to pupae…” This is hard to infer from Table 1 as you haven’t reported any Q10 values, also the data presented in Figure 3 makes me believe that thermal sensitivity of metabolic rate is much greater among adults than larvae or pupae.
• Line 273-275: “For every 1°C increase in temperature, the was a 0.056 ml h-1 increase in V̇CO2 in adults, 0.037 ml h-1 increase in larvae and 0.007 ml h-1 increase in pupae.” Providing a rate increase like this suggest that metabolic rate increases linearly with temperature, which isn’t the case. Generally Q10 is a better descriptor as it considers the non-linear effects of temperature.
• Results, generally: You include body mass as a covariate but never describing whether it had a significant effect on metabolic rate or not (until the discussion). If it did, consider plotting mass-specific metabolic rates (ml h-1 g-1) in your figures.
• Results, generally: Why are your estimates of overall VCO2 (Figure 1, Figure 5) much lower than those reported for your temperature-VCO2 curves (Figure 3, Table 1)? This is confusing: please clarify how these measures of VCO2 were calculated, if you mean for them both to be measures of metabolic rate (see comment above for Methods), and clarify why they are different.

Discussion

• Lines 357-363: This is introductory material, not material for a discussion. Also, some of this information is clearly paraphrased from sentences in the introduction.
• Lines 368: “Body fat tissues” should be “Fat body…”, if you are referring to fat body in insects.
• Lines 377: “…but in our case, size was not significant.” This result is not described in the results section.
• Lines 385-386: “only a few studies looked at that at different temperature points” I disagree with this statement. The effect of temperature on metabolic rate (including in insects at different temperature points!) has been looked at by many, many studies.
• Line 389: “Table 1a”. There is no Table 1a. Suggest ‘Table 1”?

·

Basic reporting

The manuscript is written fairly clearly and the literature referenced is relevant and adequate. Some changes to the English are needed in places. Article structure is good, although the figures could be improved. The manuscript is self-contained with relevant results and hypotheses. Specific comments are provided in the general comments to authors.

Experimental design

The research questions are well defined and interesting. Measurements are done well and most aspects of the methods, especially the respirometry, are reported with great details. I have concerns about the adequacy or reasons for the low sample sizes and the limitations that this has for the study, which are detailed in specific comments that are provided in the general comments to authors.

Validity of the findings

The findings are clear enough, but are undermined by low sample sizes and low power that reduces the robustness of the findings. Figures and in text results could be more clearly aligned if means were shown together with the boxplots. The data provided demonstrate another finding that mass loss during thermolimit respirometry occurs differently for the different life stages - this in context of water loss and CTmax could be reported and discussed. The limitations of the findings because of low sample sizes, in my opinion, should at least be acknowledged.

Additional comments

This study investigates the upper temperature limits (CTmax) and metabolic rates (VCO2) of different life stages of a pest insect, Helicoverpa punctigera. The authors found differences in the life stages for both CTmax and VCO2, but no differences between males and females. The manuscript addresses questions that I think are important to ask in the thermal tolerance space – differences with sex and life-history, and of pest species. The manuscript was an interesting read and the methods to measure thermolimit respirometry and detect activity are really neat. However, there are many points in the manuscript that I think need clarification or better discussion and there are two significant points that are the sample size and the lack of connection of the CTmax and VCO2 rates with the change in mass during the measurement. I hope that my suggestions and questions are useful for the authors when revising their manuscript.

1) Sample sizes are very low and diminish the conviction with which these results can be stated. It is fortunate in a way that the results are quite clear despite the low power, but the potential for making false positive errors is reasonably high. For instance, even one or two more individuals could change the statistical significance of the results in some cases (e.g. the life stage x sex interaction with CTmax in mean differences reported in text vs boxplots in the corresponding figure illustrates how tenuous this is). This is a shame because I think there is great potential value in the questions that this study asks and the data that has been gathered.

2) From the data provided, you can calculate mass loss during thermolimit respirometry, which differs substantially among life stages and is also relatively high (a quick calculation and glance at the numbers shows ranges of mass loss as ~13-21% in adults, ~20-24% in larvae, and ~1-3% in pupae). Although this leans into speculation, one could interpret this in a few ways. E.g., as passive differences due to changes in spiracle function/structure across life stages and as a result of different rates of evaporative water loss due to respiration. Or it could be simply due to the respirometry methods where the very dry (Drierite-scrubbed) air increases water loss potential in the insects. Was water measured by the Li7000 during measurement? However, could it also be an active process of using evaporative water loss to cool down at high temperatures and could therefore be increasing CTmax? This is certainly something that occurs in plants in response to high temperatures is there is sufficient water available to lose and I suspect the larvae stage has the greatest water resource availability (from constant foraging) to potentially employ this strategy. I think the change in mass data could be worth thinking about more, including, and discussing in this context, as it adds potentially useful explanatory information about differences in CTmax and VCO2 across life stages.

Specific points:

L38: the word ‘impacts’ is not needed as ‘impact’ already occurs earlier in the sentence

L59-60: I find this counterintuitive - a slow temperature ramp leading to heat shock – the ‘shock’ is likely to be stronger if the organism is abruptly introduced to a high temperature or rapidly increased to that temperature. The other process that can occur during a slow ramp is heat hardening or short-term physiological adjustments (e.g. induction of heat shock proteins) that protect cells from heat damage and might increase CTmax).

L64-67: but apart from this mention where you say that “thermal performance curves must be used”, you don’t then use or even discuss thermal performance curves again, so doesn’t this undermine your approach? If it is really necessary, then it should have probably been done – whereas I think there is value in CTmax and thermolimit repirometry without it, so consider changing the wording around this to be less self-critical.

L72: although it makes good sense and it may or can increase the range of environments an organism can function or survive in, “plasticity is often highly adaptive” is a commonly held assumption but one that needs further substantiation. For example, thermal plasticity in animals and plants has some but relatively limited support for having direct (positive) effects on fitness (necessary for it to be adaptive in an evolutionary sense) - see Arnold et al 2019 Phil Trans B 10.1098/rstb.2018.0185 for a review on the topic.

L68-79: as with the comment above, this paragraph would benefit from a very clear definition of what is meant here by “physiological plasticity” – it starts with thermal phenotypic plasticity and then goes into physiological plasticity – but are they the same in the context of the paper? Often people refer to plasticity but aren’t specific about what trait or aspect of biology is plastic, and this is the key point: there are many aspects of an organism’s biology that can be plastic, but only some of which will likely matter in context of response to changes in their environment. I think it is worth clarifying your use of terminology and definitions, including physiological plasticity, thermal variation, and fitness to be more specific and appropriate to the context of your study.

L85: change ‘modify’ to ‘to be modified’ and L86: ‘narrow’ to ‘narrowed’

L105-107: phrasing is confusing

L108-110: could combine these two sentences to remove redundancy

L117-121: specific dynamic action should probably be referred to in the first sentence of this paragraph where it is being defined.

L127: not sure about use of the phrase ‘The unique CO2 output’

L142-143: this sentence and reference to work on metabolic rate responses to insecticides doesn’t inform the reader of the results of what happens to metabolic rates with exposure or more important why this is relevant other than being a recent paper from the lab group.

L144-146 is redundant with L154-155.

L148-149: very cool feature that sets the temperature and exposure time relevant to body size and surface area!

L159: might be worth pointing out here or earlier when burrowing is mentioned that the reason for this is that the soil the pupae is burrowed in is far more thermally stable than leaves.

L209-210: 0.25˚C/min * 140min = 35˚C increase, but the ramp was only 25-55˚C (30˚C is the range), so some number here must be off and should be corrected, or if the temperature is held constant at the hottest temperature or something like that, it should be mentioned.

L224-225: only 5, 10, 10 individuals for each life stage is very few data points… From experience I completely understand that measuring VCO2 with flow-through respirometry and using Li7000s can be very slow throughput (~ 1 individual / 2-3 hours), but even so, these sample sizes (especially for a pest insect species that is reared in the lab) are very small and could be problematically so for analyses of the data and their subsequent interpretation, so justification for so few individuals is really needed.

L251-255: please clarify which measurement of body mass was used (before or after measurement are provided in the dataset). As in my major suggestions, I think another analysis of the change in mass from thermolimit respirometry would be useful and simple to include in this study since the data is already available. In a broad sense, there is good reason to include body mass in the models for metabolic rate (allometry: Darveau et al 2002 Nature 10.1038/417166a; White et al 2019 Nat Ecol Evol 10.1038/s41559-019-0839-9). However, I think it is also worth including body mass in the final model of CTmax and reporting the lack of relationship properly rather than saying it wasn’t significant and therefore excluded. Recent studies have investigated the patterns of CTmax and body size across species with some mixed findings that could be worth discussing (e.g. Leiva et al 2019 Phil Trans B 10.1098/rstb.2019.0035; Agudelo-Cantero & Navas 2020 J Therm Biol 10.1016/j.jtherbio.2019.03.010; Rubalcaba & Olalla-Tarraga 2020 10.1111/1365-2656.13181

Results throughout: please supply degrees of freedom with t values.

L285-287: these results do not seem to align with the boxplots provided in Figure 4/3 (page 44). This is probably because of the boxplots showing median not mean, so if possible, please provide a point for the mean of the category overlayed with the boxplot.

L324: an additional limitation of the study is unfortunately the sample size

L342-343: Very high temperatures such as those nearing CTmax would only be beneficial if food, and most importantly water, are not limiting factors – the dependence on water (in context with up to 24% mass loss during <3h of respirometry at high temperatures) needs to be discussed since such high temperatures are also frequently associated with dry conditions. The paragraph of L372-384 touches on water/dry conditions a little, but more could be said.

L398: ‘of’ should be ‘on’

L400-401: sorry, I found this sentence confusing

L416: did you observe discontinuous gas exchange patterns during respirometry or did the ramping temperature prevent discernible patterns from being clearly detected?

Figure 1, 2, and 3 (but labelled figure 4) and Table 1: the order that the life stages appear in as categories along the x-axis of figures and rows in the table appearing in a non-logical life stage order (adult, larvae, pupae or adult, pupae) – please change to larvae, pupae, adult to reflect the logical order of life stages.

Figure 3 (page 46) is not even mentioned in the manuscript. Please make sure it’s referred to and discussed (or leave out or include in a supplement)

Figure 3 or 4 (appears on page 44): the post-hoc letters are either misplaced or the median is very different to the mean report in text (L285-287). From left to right they are a,a,b,b but really look like they should be a,b,a,b from the boxplots because adult F and pupae F are very similar but adult F and adult M are not (yet these adult M/F are apparently not different in terms of means in text). Please check and correct if need be.

Table 1: clarify whether the 95% CL is SD, SE or 95% confidence intervals – all three are mentioned across the methods, table caption and table itself, so it is not obvious what values are actually shown.

Reviewer 3 ·

Basic reporting

I think the framework of the paper could be tightened significantly. The introduction is not particularly well organized, and it makes it hard for the reader (or reviewer) to really understand WHY the study is important, what the main gaps in knowledge are, and especially the theoretical basis of some of the predictions and expectations. I also think they authors should pay special attention to grammatical mistakes (there are many and I have pointed some of them out) and sentence structure. I would like the authors to do a much more thorough search in the literature for work done by numerous scientists addressing many of these issues, instead of citing the same papers over and over again.

Experimental design

No comment.

Validity of the findings

No comment.

Additional comments

The authors report the results of their experiments investigating how thermal physiology (specifically metabolic rate and critical thermal maxima) vary across ontogeny in a moth. I think the topic of the study is important; indeed we know very little about why insect populations are plummeting and certain thermal physiological traits are largely unknown for most insect adults, let alone their juvenile stages. The paper seems well-suited to Peer J’s scope and broad audience. Having said this, I do have a number of issues with the manuscript that I detail below. I mean for these comments to be helpful as the authors revise their manuscript.

MAJOR COMMENTS:

OVERALL: I think the framework of the paper could be tightened significantly. The introduction is not particularly well organized, and it makes it hard for the reader (or reviewer) to really understand WHY the study is important, what the main gaps in knowledge are, and especially the theoretical basis of some of the predictions and expectations. I also think they authors should pay special attention to grammatical mistakes (there are many and I have pointed some of them out) and sentence structure. I would like the authors to do a much more thorough search in the literature for work done by numerous scientists addressing many of these issues, instead of citing the same papers over and over again. Here are just a few papers to pay attention to:

Addo-Bediako, Kimberley Sheldon, Arthur Woods, Sylvain Pincebourde, Raymond Huey, Alisha Shah, Michael Dillon, Sarah Diamond

Addo-Bediako, A., Chown, S. L., & Gaston, K. J. (2000). Thermal tolerance, climatic variability and latitude. Proceedings of the Royal Society of London. Series B: Biological Sciences, 267(1445), 739-745.

Deutsch, C. A., Tewksbury, J. J., Huey, R. B., Sheldon, K. S., Ghalambor, C. K., Haak, D. C., & Martin, P. R. (2008). Impacts of climate warming on terrestrial ectotherms across latitude. Proceedings of the National Academy of Sciences, 105(18), 6668-6672.

Sheldon, K. S., & Tewksbury, J. J. (2014). The impact of seasonality in temperature on thermal tolerance and elevational range size. Ecology, 95(8), 2134-2143.

Diamond, S. E., Nichols, L. M., McCoy, N., Hirsch, C., Pelini, S. L., Sanders, N. J., ... & Dunn, R. R. (2012). A physiological trait‐based approach to predicting the responses of species to experimental climate warming. Ecology, 93(11), 2305-2312.

Shah, A. A., Woods, H. A., Havird, J. C., Encalada, A. C., Flecker, A. S., Funk, W. C., ... & Ghalambor, C. K. (2021). Temperature dependence of metabolic rate in tropical and temperate aquatic insects: Support for the Climate Variability Hypothesis in mayflies but not stoneflies. Global change biology, 27(2), 297-311.

Pincebourde, S., Dillon, M. E., & Woods, H. A. Body size determines the thermal coupling between insects and plant surfaces. Functional Ecology.

These are only a handful of papers that I think have excellent theoretical background and could help the authors revise the framework more succinctly. There are of course many, many more. I continue to detail my comments below by section.

INTRODUCTION:

1) The introduction isn’t as well organized as it could be. In some cases, the beginning of a paragraph does not set up the information in the paragraph. A general rule of thumb is that the first sentence of the paragraph should allude to the theme of that paragraph- also try to stick with one idea per paragraph. For example, in the paragraph starting on Ln 53: The first sentence seems to suggest you will talk about different life stages and why they might differ in CTmax. However, the rest of the paragraph is about how CTmax values are sensitive to methodology. In addition, please pay attention to the transitions between paragraphs. Each paragraph should have a transitional sentence that prepares the reader for new information from the preceding paragraph. As it stands, many of the paragraphs read like separate, disparate thoughts, one after the other. An example of this is on the paragraph ending at Ln 67 and the one beginning on Ln 68. Perhaps mention plasticity in the previous paragraph so the following one doesn’t seem to be a completely new topic.

2) I feel that introduction is lacking a clear and robust discussion on why different life stages should differ in thermal tolerance (and corresponding citations).

3) Lns 37-38: What exactly is meant by “population dynamics” in this context? Population dynamics typically refers to the changes in population size as a function of recruitment of young, death, and migration. I am struggling to see the connection between studying the thermal limits, metabolic rates and population dynamics in the paper, because you are not measuring those dynamics. I would like to see a clear discussion of these ideas to set up the introduction more throughly. The importance of physiology on population dynamics is never mentioned in the Discussion either.

4) Lns 101- 102: I read this sentence as if CTmax is the only determinant of survival for a small insect at extreme temperatures. I disagree. What about biochemical shifts that are temperature-dependant and happen long before CTmax is achieved (see Hotchachka and Somero’s work). Similarly, what about the danger of desiccation, which is linked with temperature tolerance? I caution the authors about writing sentences in such absolutes. All thermal ecologists and physiologists agree that these traits are complex and we still don’t know exactly how they work or affect organisms.

5) Lns 103 - 137: there is certainly a lot of theory about how metabolic rate should change as a function of temperature. There are also a variety of metabolic rate measurements, e.g. resting or basal metabolic rate (which, I think is what you measured). Active MR measures other aspects of an animal’s energy budget. At a minimum, I would like to see a more clear explanation of what type of MR was measured (resting/active), why it was measured, and why you might expect differences to exist between the life stages. I understand that some of the information is already in the manuscript, but a more concise paragraph with these details may help the flow of the introduction.

METHODS:

Ln 172; What temperature were the eggs kept at?

Ln 177: I did not know you could clean lepidopteran eggs this way!

Ln 218: Should this whole sentence be bolded instead of just parts of it?

I think you used a clever way of standardizing CTmax measurements across life stages. Nice.

Note that word “data” is plural so please write, “data are”, “data were”, not “data is” or data “was”.

Ln 246: do you mean “life stages”?

If you are not dealing with mass with CTmax, you could provide the F statistic and p value for CTmax regressed over mass to show it was non-significant.
I am surprised that you get such strong results with such a small sample size. Am I correct that you only measured 5 larvae?

Lns 16-17: I would say “upper thermal limit”. Tolerance is typically a range, whereas a limit is a set point(s), which is what you are referring to in your paper.

Ln 37: delete “moving”. Do you mean shifting the dynamics? or simply the dynamics (which implies variation and change).

Ln 38: “impact” is used twice in the same sentence. Revise.

Ln 45-46: Limits -> limit. Missing “ are” before commonly.

Ln 99: Should say “pupates”

Ln 102: Comma after e.g.

Ln 128-128: Should say “performs”

Lns 140-141: Decide between “thermophysiology” and “thermal physiology”. I personally prefer the latter and believe it is grammatically more accurate. Regardless, pick one and be consistent.

Ln 255: I think you mean Tukey.

RESULTS:

The figure numbers do not match the figure references in the results. Please go through it carefully. For example, you never mention Figure 3 in the main text.

Figures 3 & 4 caption numbers are switched.

I would like to see sample sizes somewhere in the graphs, captions, or in a separate table.

Figure 3: Are pupal CTmax values actually different from the adult values? If they were not part of the analysis, then why are they shown together on the same plot?

DISCUSSION:

Lns: 296-297: You could cite many other much more recent studies on global insect declines.

Ln 317: change to “evolutionarily”

Ln 321-324: I am not sure I understand your explanation. Are you saying that because there is low genetic diversity among lab-reared and wild populations, you expect thermal physiology to be similar between the two groups? I don’t think this is necessarily true. Genetics is not all that accounts for differences in physiology, as you have mentioned elsewhere in your paper. Think about thermal history, plasticity, trade-offs, etc.

Ln 335-337: To me it seems logical that you would expect thermal physiological differences between adult and larval stages. On one hand, larvae are so much more sedentary and likely have a much harder time thermoregulating via behavior, so one could argue they should have a higher tolerance to heat stress. Adults, on the other hand, are far more mobile and could be better-suited to seeking preferred temperatures. This might result in the evolution of lower heat tolerance in adults compared to larvae.

Lns 346-348: Can you say this a little more clearly? I don’t understand how “microclimatic conditions… are higher”. Do you mean that the maximum temperatures are higher? If so, how is that possible? If you have data to back up your statement, I think it would be good to put it here.

Lns 354-355: Indeed, your data suggest that males and females do not differ in their upper thermal limits during short term exposure. But what about long-term exposure or long-term effects on fecundity?

---

## Round 0.2 · Minor Revisions

Thank you for your revised manuscript. The Reviewer and I have a few comments. Please address the Reviewer's specific comment on the outlier in Figure 3.

In addition I have some minor comments:

1) Can the introduction and discussion be shortened for clarity? Can the paragraphs beginning on Lines 106 and 112 be combined? Is the information on Lines 139-144 necessary in the introduction? Perhaps it could be used to support your analyses in your discussion?

2) L163-165: The way this sentence is written is confusing is the cage 50 ml, or is that the size of the container containing sugar water? I suspect the latter, but it doesn't read that way.

3) L295-297: The way these results are written it is difficult to interpret. Particularly, what statistics correspond to what life stage. Please clarify.


Minor:

L17: remove "on"

L51-52, 68-69: Please give the scientific names along with the common names.

L107: Change "gender" to "sex"

L121: Change "use" to "used"

L221 and 229: use "10" instead of ten after "n = ..."

L295: There is a hanging "and pupae" please remove or clarify.

L307: Change "adult" to "adults"

L320: Change "implies" to "imply"

L323: Remove "being"

L348: insert "are" after "larvae"

L365: Give the scientific name for the species of bumble bee.

L392: Remove "the" before "more fat"

L393-395: This is a little confusing. Please revise for clarity.

L448-449: This sentence appears to be hanging. What do the results suggest about the performance and survival of the different life stages?

·

Basic reporting

no further comment on original

Experimental design

no further comment on original

Validity of the findings

no further comment on original

Additional comments

The authors have extensively revised their manuscript in response to the numerous comments from all reviewers. The revised version has been improved greatly. Responses to some questions raised by reviewers were not elaborated on much, but they have at least made a substantial effort to revise the manuscript text, figures, and supporting information in line with the reviewer comments.

I have some new minor comments:

L89-92: this sentence presents three or four very disjointed ideas in it and it itself seems out of place. The content is fine, but it needs to be separated into smaller sentences to make sure the links among them are logical.

Figure 3: the outlying Q10 35-45˚C value for pupae that is apparently above 10 is surely unrealistic when the remaining pupae estimates are around 1-2… please check the data and even if it appears technically sound, consider carefully whether such a value is biologically realistic. I really doubt it is.


Also, a few small editorial suggestions:

L39: change ‘an’ to ‘that an’

L70: change ‘ground surface as well as on plant surface’ to ‘ground and plant surfaces’

L71: change ‘the many microclimatic conditions including’ to ‘substantial microclimate variability, including’

L370: change ‘have’ to ‘has’

---

## Round 0.3 · Minor Revisions

Thank you for your revisions. Your answer to the previous review comment of "The figure is technically sound and we argue that the value is biologically relevant." is insufficient for me to recommend this article be accepted at this point. Please address this comment in more detail in the rebuttal and/or manuscript, why is this outlier valid and biologically relevant?

---

## Round 0.4 · accepted · Accept

Thank you for addressing my previous comments.